# Structures of Coxsackievirus A10 unveil the molecular mechanisms of receptor binding and viral uncoating

Ling Zhu[1], Yao Sun[1], Jinyan Fan[2], Bin Zhu[3], Lei Cao[1], Qiang Gao[4], Yanjun Zhang[5], Hongrong Liu[3], Zihe Rao[1,6] & Xiangxi Wang [1]

Coxsackievirus A10 (CVA10), a human type-A *Enterovirus* (HEV-A), can cause diseases ranging from hand-foot-and-mouth disease to polio-myelitis-like disease. CVA10, together with some other HEV-As, utilizing the molecule KREMEN1 as an entry receptor, constitutes a KREMEN1-dependent subgroup within HEV-As. Currently, there is no vaccine or antiviral therapy available for treating diseases caused by CVA10. The atomic-resolution structure of the CVA10 virion, which is within the KREMEN1-dependent subgroup, shows significant conformational differences in the putative receptor binding sites and serotype-specific epitopes, when compared to the SCARB2-dependent subgroup of HEV-A, such as EV71, highlighting specific differences between the sub-groups. We also report two expanded structures of CVA10, an empty particle and uncoating intermediate at atomic resolution, as well as a medium-resolution genome structure reconstructed using a symmetry-mismatch method. Structural comparisons coupled with previous results, reveal an ordered signal transmission process for enterovirus uncoating, converting exo-genetic receptor-attachment inputs into a generic RNA release mechanism.

[1] National Laboratory of Macromolecules, Institute of Biophysics, Chinese Academy of Sciences, Beijing 100101, China. [2] Beijing Productivity Center, Major Project Department, Beijing 100088, China. [3] College of Physics and Information Science, Synergetic Innovation Center for Quantum Effects and Applications, Key Laboratory of Low-dimensional Quantum Structures, and Quantum Control of the Ministry of Education, Hunan Normal University, Changsha 410081, China. [4] Sinovac Biotech Co., Ltd, Beijing 100085, China. [5] Zhejiang Provincial Center for Disease Control and Prevention, Hangzhou 310051, China. [6] Laboratory of Structural Biology, Tsinghua University, Beijing 100084, China. These authors contributed equally: Ling Zhu, Yao Sun, Jinyan Fan. Correspondence and requests for materials should be addressed to L.Z. (email: lingzhu@ibp.ac.cn) or to Y.Z. (email: yjzhang@cdc.zj.cn) or to Z.R. (email: raozh@xtal.tsinghua.edu.cn) or to X.W. (email: xiangxi@ibp.ac.cn)

The *Enterovirus* (EV) genus, one of the most populous within the *Picornaviridae* family, is comprised of more than 100 serotypes afflicting millions of people worldwide annually[1]. Coxsackievirus A10 (CVA10), belonging to the human type-A *Enterovirus* (HEV-A) subgroup, used to be associated mainly with herpangina, but in recent years has been increasingly reported to co-circulate with other members of the genus such as enterovirus 71 (EV71), CVA16 and CVA6 causing hand, foot, and mouth disease (HFMD) outbreaks in Asia, Europe, and North America[2,3]. Furthermore, CVA10 infections have also been shown to cause severe clinical symptoms such as aseptic meningitis in infected children, possibly due to genomic recombination events within HEV-As[4]. Currently, there are no approved vaccines or antiviral therapies available for treating infections caused by CVA10, and existing EV71 vaccines do not cross-protect against CVA10 infections[5]. Therefore, an in-depth understanding of CVA10 should be useful in providing guidance for the rational design of novel and effective multivalent vaccines for protection against EVs.

Like most members of the picornavirus family, HEV-As comprise 60 copies of viral proteins VP1, VP2, VP3, and VP4 arranged in pseudo $T = 3$ symmetry, encasing a positive-sense single-stranded RNA genome. While VP1, VP2, and VP3 are exposed on the surface of the virus, the smaller VP4s line the interior of the virus. As part of a normal picornavirus infection, natural empty particles devoid of RNA are also often formed, in which the final protein cleavage from VP0 to VP2 and VP4 does not occur[6,7]. To initiate infection, EVs undergo conformational changes that lead to the delivery of their genome into the host cytosol. A depression encircling each fivefold axis referred to as "canyon" is located on the surface of EVs. A hydrophobic pocket within VP1 is located below the canyon base and normally contains a fatty acid molecule[7,8]. Upon receptor binding, the mature virus releases the fatty acid molecule, leading to a cascade of structural changes, including the loss of VP1 N-terminus and VP4, resulting in the formation of an expanded particle, namely, A-particle or uncoating intermediate[9,10]. This is followed by delivery of the viral genome to the cytoplasm of the target cell via engagement with the endosomal membranes, leaving an empty particle or B-particle[11].

Although atomic structures of three members of HEV-A, EV71, CVA16, and recent CVA6 (expanded empty particle and A-particle, but no structure for mature virion) have been reported[7,12–14], there is no high resolution structural information available for CVA10. Intriguingly, although CVA10 shares ~69% amino-acid sequence identity with EV71 and CVA16, it utilizes a different receptor, KREMEN1, for entry into host cells[15], suggesting CVA10 might differ in its mode of entry into host cells or perhaps even in its local structural features that are involved in receptor recognition. CVA10 forms a major KREMEN1-dependent subgroup of HEV-As for which no structural information (on mature virus) is available, together with two other sub-groups of HEV-As, SCARB2-dependent, and X-receptor (unidentified)-dependent sub-groups, making up the HEV-A family[15,16].

Here, we report the atomic model of CVA10 mature virus built from a 3.0-Å resolution cryo-EM map. The overall structure of CVA10 resembles the structures reported for SCARB2-dependent HEV-As, while significant differences exist on both the outer and inner capsid surfaces, including the putative receptor-binding sites and serotype-specific epitopes, which is in line with the specific differences between the virus types. We also determined cryo-EM structures of CVA10 empty particle and A-particle at 2.8 and 2.7 Å resolution, respectively, that both closely resemble those of the expanded EV uncoating intermediates previously visualized by cryo-EM[13,17,18]. The structures of CVA10 reported in this study, which include lower resolution asymmetric information on the interaction of the genome with the A-particle, coupled with expanded particle structures from our previous investigations of EV71 and CVA16, allow us to propose a detailed molecular mechanism for the early stages of HEV uncoating.

## Results

**Structure determination.** CVA10 was isolated from a patient from Zhejiang, China, and cultivated in Vero cells and purified by centrifugation, linear and discontinuous sucrose gradient ultra-centrifugation and ultrafiltration. Three types of particles were separated, two containing significant amounts of viral RNA with distinguishable sedimentation coefficients (~160 s and ~130 s, characterized as mature virus and A-particle, respectively) and one being empty inside with the smallest sedimentation coefficient (~80 s, characterized as empty particle) (Supplementary Fig. 1). Cryo-EM micrographs of purified CVA10 mature virus, empty- and A-particles were recorded using an FEI Titan Krios electron microscope equipped with a Gatan K2 Summit detector (Supplementary Fig. 2). A total of 4586, 22,725, and 21,456 particles were used to reconstruct the structures of CVA10 mature virus, empty- and A-particles with icosahedral symmetry imposed by single-particle techniques using Relion[19]. The resolutions of the final maps for CVA10 mature virus, empty- and A-particles are 3.0, 2.7, and 2.8 Å, respectively using the "gold" standard Fourier shell correlation (FSC) = 0.143 criterion[20] (Supplementary Fig. 3). The backbone of the polypeptide, as well as many side chains, was clearly defined for most of the capsid, allowing atomic models of the majority of the capsid proteins for three types of CVA10 particles to be manually built in COOT[21] (Fig. 1a–e). The models were refined and validated using standard X-ray crystallographic metrics (Supplementary Table. 1).

**Structure of the mature virus.** Except for a few disordered residues in VP4 (residues 1–18) and VP2 (residues 1–9) (Fig. 1e), the mature CVA10 is well ordered. The overall structure of this viral particle is similar to those of EV71 and CVA16 (Fig. 2a). The capsid proteins VP1, VP2, and VP3 adopt the classical eight-stranded anti-parallel β-barrel configuration and follow the expected pseudo $T = 3$ symmetry, where VP1 surrounds the fivefold axes and VP2 and VP3 alternate about the two- and threefold axes (Figs. 1e and 2b). When compared to the structures of EV71 and CVA16, the external surface of CVA10 exhibits five additional continuous and "star-shaped" dense protrusions surrounding the fivefold axes. These arise as a result of the loops at the fivefold axes, in particular the VP1 BC loop, which assumes a raised configuration (Fig. 2a, b and Supplementary Fig. 4). Like other HEV-As, CVA10 also possesses the canyon, the site of receptor binding in many EVs[22–24] and the hydrophobic pocket in VP1 harboring a pocket factor beneath the canyon (Fig. 2b). However, compared to EV71 and CVA16, the raised conformation of the VP1 BC loop forming the north wall of the canyon and the refolding of the VP1 GH loop (front part of the south wall) from a short helix to a more extended loop moving away from the "star-protrusion" broaden the front portion of the canyon of CVA10 (Fig. 2b). In contrast, the VP2 EF loop (back part of the south wall) of CVA10 bears a distinct conformation and moves closer towards the "star-protrusion", narrowing the back portion of the canyon (Fig. 2b). It is quite likely that these conformational changes of the canyon and neighboring micro-environments, including VP3 BC loop, implicated in receptor and neutralizing monoclonal antibody binding[25–28], together with the alteration in electrostatic properties (Supplementary Fig. 5), determine the specificity of CVA10 for KREMEN1 during host cell entry and specify the antigenic sites of CVA10. In addition to

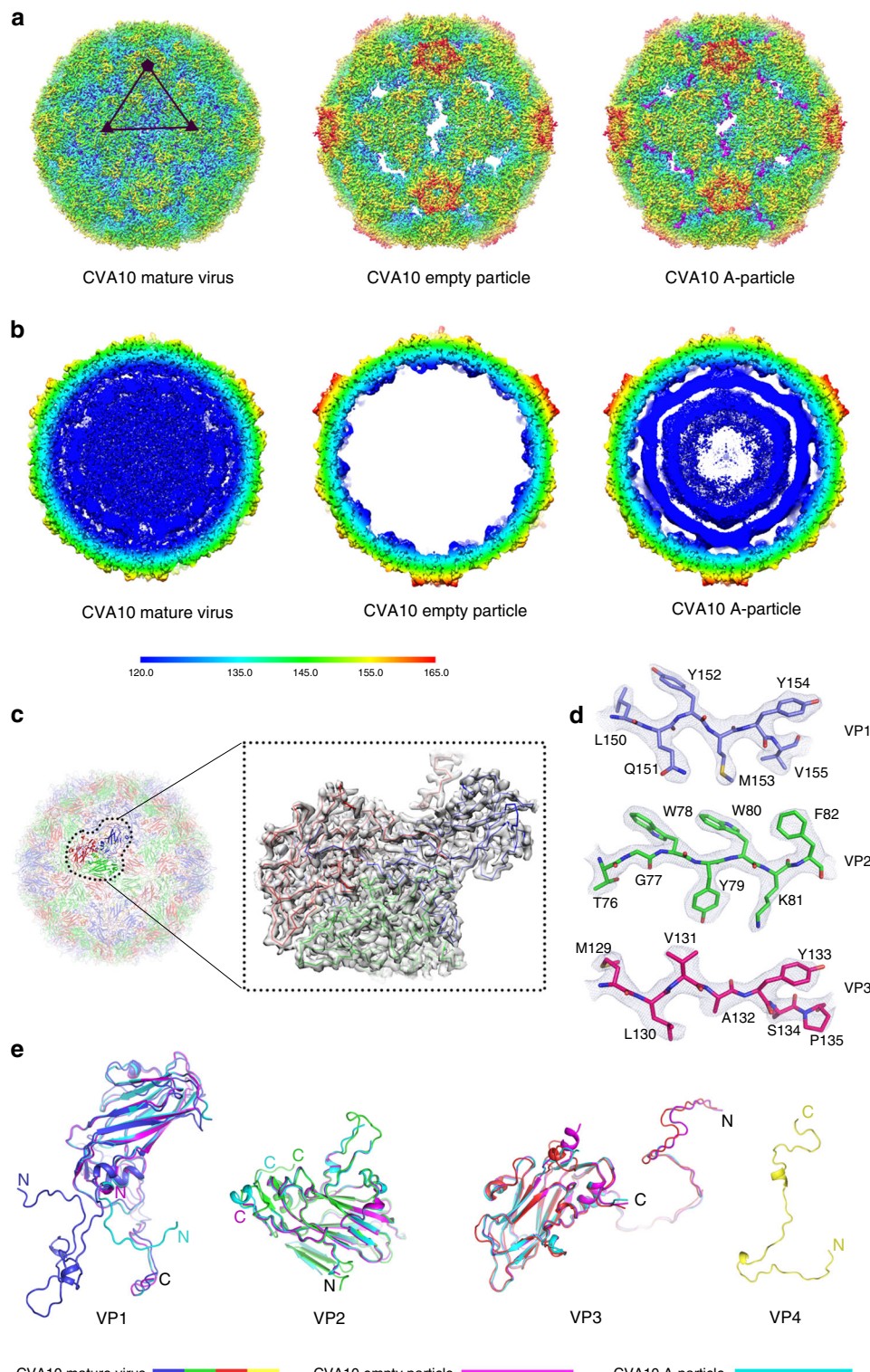

**Fig. 1** Cryo-EM structures of CVA10 mature virus, empty- and A-particles. **a** Three-dimensional reconstructions of CVA10 mature virus, empty- and A-particles viewed along the icosahedral twofold axes. The surfaces are colored radially from blue through green to red from the lowest to the highest radius. One icosahedral asymmetric unit is marked with a black triangle in CVA10 mature virus; icosahedral symmetry axes are drawn in black. The obstructed off-axis channels near the twofold axes in CVA10 A-particle are highlighted in magenta. **b** Thin slices of the central sections of the CVA10 mature virus, empty- and A-particles viewed down the threefold axes, colored radially from blue through green to red from the lowest to the highest radius. The scale bar applies to both **a** and **b**. **c** Cartoon of the mature CVA10 virus looking down an icosahedral twofold axis (left). The color scheme uses the signature colors (VP1, blue; VP2, green; VP3, red). A single icosahedral protomer is drawn more brightly and the electron density map (gray) for this protomeric unit is accentuated in inset (right). **d** Electron density maps from a section of VP1, VP2, and VP3, respectively. **e** Superimpositions of VP1, VP2, and VP3 from CVA10 mature virus, empty- and A-particles, respectively. VP1, VP2, VP3, and VP4 of CVA10 mature virus are colored using classic color scheme, VPs from CVA10 empty particle are shown in magenta, VPs of CVA10 A-particle are displayed in cyan, as shown by the color bars. N-and C-terminus of each protein are labeled in correspondingly colored letters; termini that are overlaid or near are labeled in black letters

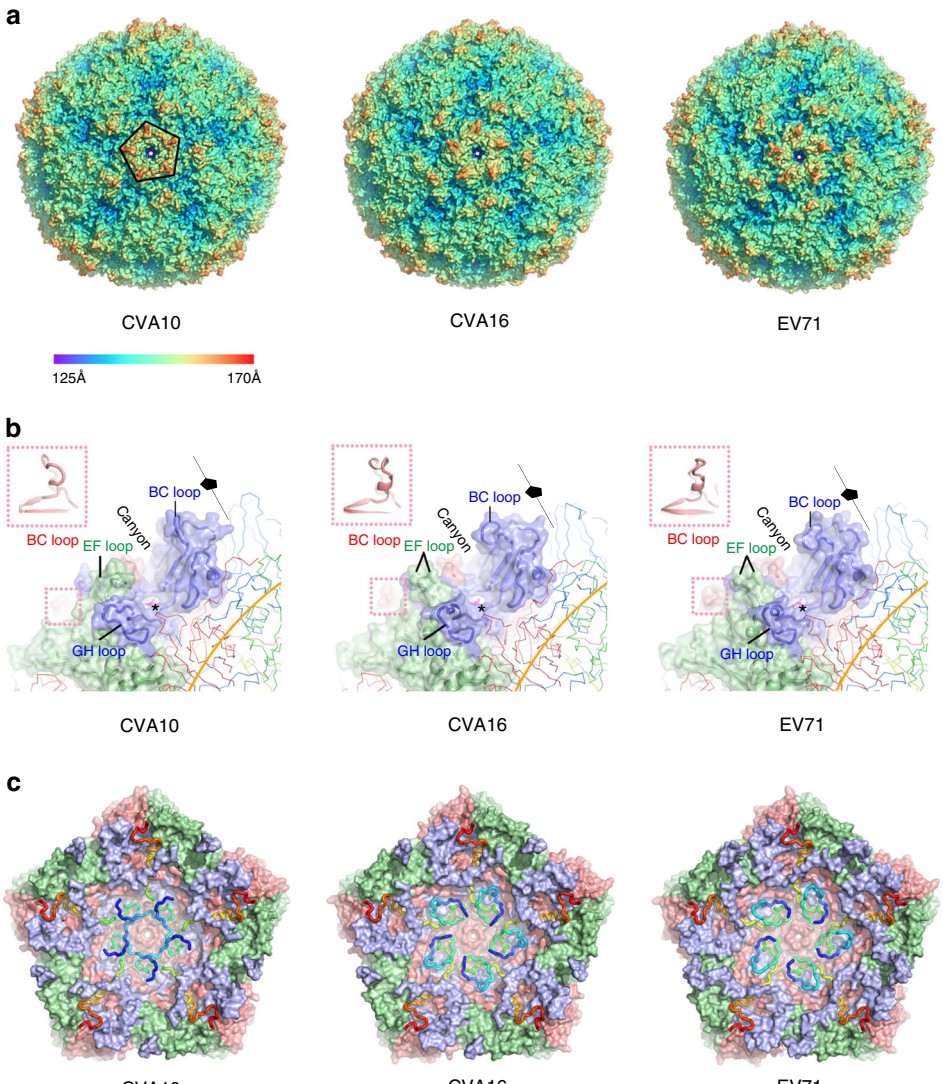

**Fig. 2** Structural comparisons of CVA10, CVA16, and EV71 mature viruses. **a** Comparisons of capsid from CVA10 mature virus with those from CVA16 and EV71 mature viruses. Capsids are colored according to their radius from blue to red as shown in the color bar. The "star-shaped" dense protrusions are marked by black lines. **b** Surfaces of the biological protomers of CVA10, CVA16, and EV71 mature viruses. The color scheme uses the signature colors (VP1, blue; VP2, green; VP3, red; VP4, yellow). Loops from a protomeric unit forming the canyon walls are labeled in correspondingly colored letters, BC loops from VP3 are highlighted in pink insets. The hydrophobic pocket is highlighted by the star symbol. **c** Rainbow ribbon representations of the VP4 structures (blue N-terminus through green and yellow to red C terminus) in CVA10, CVA16, and EV71 mature viruses. VP1 (light blue), VP2 (light green), and VP3 (salmon) structures are rendered as surfaces. A pentamer of icosahedral protomeric units from each virus is shown. The view is along the fivefold axis from the inside of the virus

the differences in the external surface, the inner surface of the capsid of CVA10 is also distinct from those of EV71 and CVA16 (Fig. 2c). Notably, in CVA10, the N-terminal portion of VP4 lies under the adjacent biological protomer rather than forming a loose spiral beneath VP1 as observed in EV71/CVA16. The most striking differences between CVA10 and EV71 as well as CVA16 are completely opposite charge distributions at the inner surface surrounding the fivefold-channel reveal a different interaction mode between the inner capsid proteins and viral RNA genome in CVA10, which is probably assembled differently to EV71/CVA16 subgroup (Supplementary Fig. 5).

**Expanded empty- and A-particles**. Compared to mature viruses, the CVA10 empty- and A-particles are markedly expanded with a ~5% increase in capsid radius (Fig. 1a, b). The expansion of these

two particles creates tectonic movements in the particle, disordering ~70,000 and ~50,000 protein atoms per particle, respectively. This results in separation of the protomeric units and opening of a major channel at the icosahedral twofold axes (Fig. 1a, b). In empty particles, there are 60 extra small channels at the base of the canyon (off-axis channels), which are plugged by VP1 and VP2 in A-particle (Fig. 1a, b). Superimposition of the empty particle onto the A-particle (protomeric unit) reveals that these two particles are very similar in structure with an r.m.s. deviation of 0.4 Å between the $C_\alpha$ atoms of corresponding residues. However, both these structures differ substantially from the structure of the mature virus (reflected in an r.m.s. deviation of 3.2 Å between corresponding $C_\alpha$ atoms). This observation further supports the proposal[7] that there are essentially two distinct configurations for EV capsids; one corresponding to the mature virion and the other to the expanded virion particle. As expected,

both expanded particles lack visible VP4 and harbor neither an hydrophobic pocket nor pocket factor in VP1 due to the movements of the polypeptide backbone in residues 229–233 at the end of the GH loop and the start of strand H, and the conformational changes of the Met229 and Phe232 side chains (Fig. 1e and Supplementary Fig. 6). Additionally, several external loops like the VP1 GH loop (residues 208–225), VP2 EF loop (residues 137–147) and VP3 GH loop (residues 181–186), that form important epitopes for neutralizing antibodies in EV71, become disordered in empty particle. Interestingly, only parts of VP1 GH loop (residues 212–223) are flexible in A-particle (Fig. 1e). These structural comparisons are consistent with the results of the immunogenic assays where the antisera elicited from inactivated CVA10 mature viruses and A-particles exhibited high and comparable neutralizing titers against CVA10, but CVA10 empty particles showed very weak immunogenicity (Supplementary Fig. 7). For many enteroviruses[7,12–14], two (mature viruses and empty particles) or three (mature viruses, A- and empty particles) predominant types of viral particles with various ratios are produced during a natural infection and the ratios of these particles can be influenced by a couple of factors, including virus strain, multiplicity of infection, cell culture conditions and cell types.

Our results provide lessons for CVA10 vaccine development: the procedures for virus production need to be optimized to decrease amounts of antigenically altered empty particles, which are likely to dilute the useful portion of a vaccine.

**Particle expansion and egress of proteins during uncoating.** Comparison of the structures of the three types of viral particles reveals that the expansion of the viral particle is accompanied by a 6.6° counterclockwise rotation of the protomeric unit at the corner of VP3 near the threefold axis as well as conformational shifts within the protomeric unit (Fig. 3a). Within the protomeric unit, the fivefold proximal end of the VP1 β-barrel moves away from the particle center by as much as 10.1 Å and VP2 shifts away from the twofold axis by 6 Å in a jackknife rotation, pivoting about the hydrophobic pocket, which is similar to the motion observed in EV71[7] (Fig. 3a). If the protomeric units from mature virus, empty- and A-particles are superposed as rigid bodies, the $C_\alpha$ positions of matched residues of empty particle and A-particle when compared to the mature virus yielded r.m.s. deviations of 1.4 and 1.5 Å, respectively (Fig. 3b), much less than that (~3.2 Å) calculated from the overlay based on the complete particles. Apart

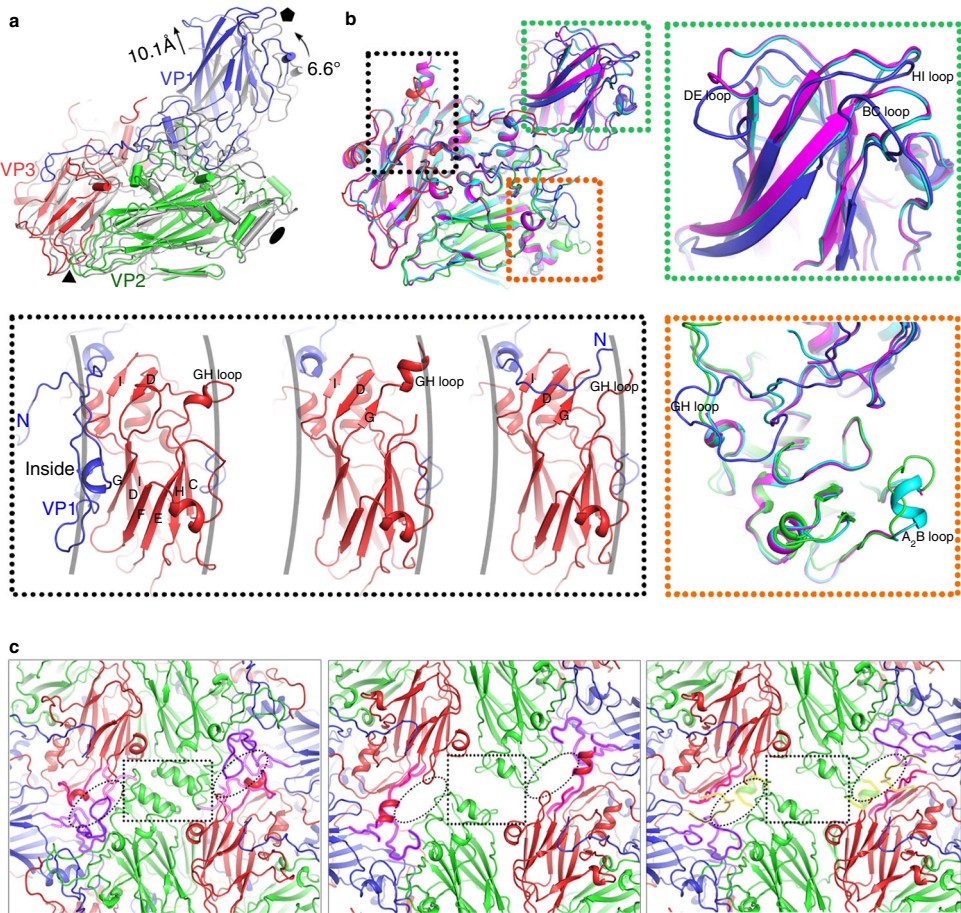

**Fig. 3** Distinctions between the CVA10 mature virus, empty- and A-particles. **a** Protomeric units shown with respect to the icosahedral axes of the particles by superposing whole particles. The mature virus is shown in gray and the A-particle is colored as in Fig. 1c. The orientation is similar to that of the bright protomer in Fig. 1c; icosahedral symmetry axes are drawn in black. **b** Superposed protomers with structural differences highlighted in colored insets. VPs from CVA10 mature virus are colored using signature scheme; VPs of CVA10 empty- and A-particles are colored in magenta and cyan, respectively; noted for the larger black inset, signature color scheme applies to CVA10 mature virus (left), CVA10 empty particle (middle), and CVA10 A-particle (right). **c** Close-up views centered at twofold axes of the atomic models from CVA10 mature virus (left), empty particle (middle), and A-particle (right). The twofold channels (dotted rectangles) are closed (left), open (middle, right). The off-axis channels (dotted ellipses) are closed (left), unobstructed (middle), and obstructed (right). VP1 GH loop and VP3 GH loop are outlined in magenta (left, middle, right), VP1 N-terminus and VP2 $A_2B$ loop are outlined in yellow (right)

from structural shifts, several loops like the BC, DE, GH, and HI loops of VP1 adopt substantially different conformations in the mature and expanded particles (Fig. 3b). Unexpectedly, the $A_2B$ loop (residues 38–50) of VP2 located between the twofold axis and the base of the canyon refolds itself from an extended loop observed in the mature virus to a short helix in the A-particle, but is flexible in the empty particle (Fig. 3b). Additionally, there are marked alterations in the VP3 GH loop, which folds into a "π" conformation in the mature virus, but forms a β-hairpin in the A-particle, and a β-hairpin plus a helix in the empty particle (Fig. 3b). The most striking differences occur at the VP1 N-terminus (residues 1–71), which lies at the inner surface of the capsid in the mature virus, and is disordered in the empty particle, but traverses the capsid terminating on the external surface at residue 62 in the A-particle (the first 61 residues of VP1 might be disordered) (Fig. 3b). A similar transition of VP1 N-terminus has been observed in CVA16[12], but the orientation of the N-terminus differs (Supplementary Fig. 8). These conformational alterations of individual capsid proteins relay a cascade of interactions, leading to large open channels at the twofold axes in expanded particles, where the separation at the twofold axes rips apart the αA helices of adjacent VP2 subunits that normally form key interactions stabilizing the mature virus (Fig. 3c). In the empty particle, refolding of the VP1 GH loop and the VP3 GH loop and partial disorder of these two loops as well as the VP2 $A_2B$ loop result in the formation of off-axis channels (Fig. 3c). It is quite likely that the rearrangements of the VP3 GH and VP2 $A_2B$ loops, together with increased volume via particle expansion, facilitate the egress of the VP1 N-terminus, obstructing the off-axis channels in A-particle (Fig. 3c), which might be triggered by receptor-attachment[7]. The A-particle transits its VP1 N-terminus (up to ~60 residues) to exit the capsid, including residues 3–22, which are likely to form an amphipathic helix[11], requiring enough space to traverse the capsid. Therefore, it seems more likely that

VP1 N-terminus exits at the twofold channel first, then moves sideways towards off-axis channel at the base of the canyon, along with rearranging the VP3 GH and VP2 $A_2B$ loops en route before being finally pinned to it. This is also in line with the cryo-EM observations which show that the externalization of internal peptide mediated by antibodies occurs at the twofold axes[29].

**Structure of the viral genome and the binding mode to capsid.** For picornaviruses, the single-stranded RNA genome is highly condensed and packed at a concentration of $> 800$ mg mL$^{-1}$ in mature virion[30], double the concentration of the dsRNA *Reoviridae*[31], forming a stable core structure (Fig. 1b). During the uncoating, the capsid must undergo ordered conformational changes, including particle expansion and loss of inner protein components, to increase internal particle volume, allowing the RNA to egress (Fig. 3c). In line with this, the genome–capsid interactions are dramatically altered and the condensing strength of counterions decreases, but the genome presents a multiple-layer structure with a ~18 Å gap between adjacent layers, including the most external layer and inner capsid shell in A-particle (Figs. 1b and 4). Although multiple X-ray crystallographic and icosahedrally reconstructed cryo-EM structures of picornaviruses show small portions of the viral RNA[32–34], given the fact that the bulk of the genome does not exhibit icosahedral symmetry, reconstruction with the imposition of icosahedral symmetry would average the densities, obscuring the genome reconstruction. To overcome this problem, we reconstructed the capsid and genome using the symmetry-mismatch reconstruction method[35] (without imposing any symmetry), being expected to become a common strategy to study asymmetric structural features in icosahedral viruses. As expected, the density for the capsid is indistinguishable from that reconstructed with imposition of icosahedral symmetry, when the two maps are low-pass

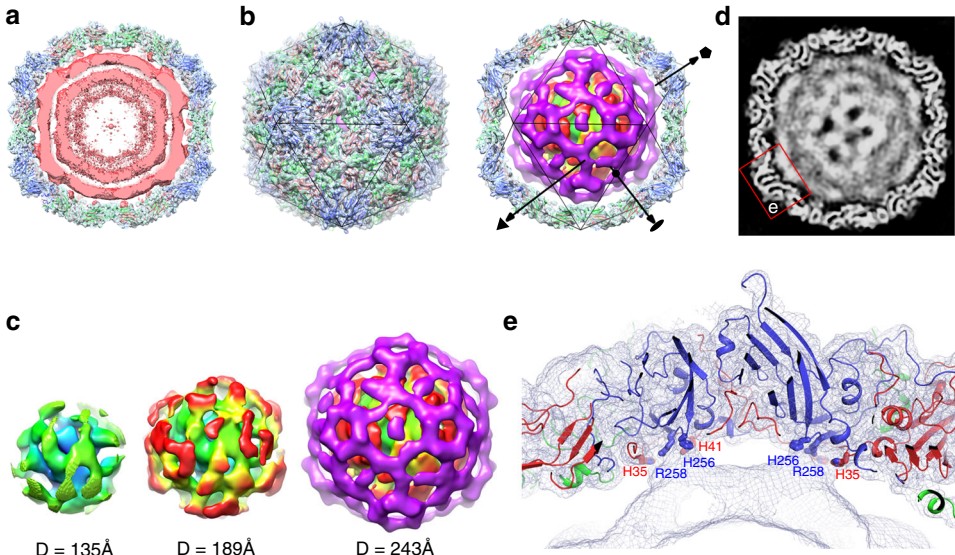

**Fig. 4** Structure of CVA10 A-particle genome and its contacts with the capsid. **a** Central slab of the CVA10 A-particle density map of 22 Å thickness reconstructed by applying icosahedral symmetry and corresponding atomic model (ribbon diagrams) viewed down an icosahedral twofold axis. The densities for capsid shell and genome are colored in light blue and pink-red, respectively. Capsid–RNA interactions are observed beneath the VP1 and VP3 subunits. **b** The symmetry-mismatch reconstructed capsid structure of CVA10 A-particle (left) and a central slab of 22 Å thickness of the CVA10 A-particle capsid structure together with the structure of genomic RNA (right). The threefold axis, twofold axis, and fivefold axis of symmetry are indicated by a triangle, an ellipse and a pentagon, respectively. **c** Structures of genomic RNA layers at different diameters viewed along the twofold axes, and are colored radially from blue, green through yellow, orange to magenta. **d** Central slice of the CVA10 A-particle structure showing the capsid and genomic RNA. Close-up of inset **e** is shown. **e** Slab of 22 Å thickness of CVA10 A-particle at the fivefold axis with the external surface of the capsid facing upwards. VP1 (blue), VP2 (green), VP3 (red) are shown. Contacts between the genomic RNA and capsid are mediated by VP1 (H256, R258, labeled in blue) and VP3 (H35, H41, labeled in red)

filtered to 5 Å (the symmetry-mismatch reconstruction yields a ~5 Å map for capsid shell) (Fig. 4a, b). The structure (at ~ 15 Å resolution) of genome RNA is of spherical outline, and is composed of regularly distributed layers that are possibly formed by discontinuous dsRNA fragments or pseudoknots and exhibits pseudo-fivefold symmetric organization at the outermost layer (Fig. 4b, c and Supplementary Fig. 9). The specificity of the interaction between the inner capsid and the genome RNA reveals that the contacts from capsid-RNA might partially follow the symmetry of the capsid, which is similar to the observations in human parecho, Ljungan and Seneca Valley viruses (Supplementary Fig. 10). However, more extensive and continuous capsid–RNA interactions are identified in CVA10 A-particle (Supplementary Fig. 10). Additionally, a number of links between the inner capsid and genome density are observed, one of which is contributed by the residues H256, R258 of VP1 and H35, H41 of VP3, occurring beneath the base of the canyon, adjacent to the point where VP1 N-terminus exits the capsid as well (Fig. 4d, e). These structural features suggest an ordered signal transmission process for enterovirus uncoating, converting exo-genetic receptor-attachment inputs to a generic RNA release mechanism.

## Discussion

How the RNA genome is productively released from picornaviruses has long been the subject of speculation[36–38], and although the details vary, the process seems fundamentally similar for all EVs. Here, we provide models for CVA10 mature virus, empty- and A-particles at atomic resolution, and establish details of the events covering a staged exit of the VP1 N-terminus and ordered structural reorganizations of VP1 GH, VP2 A₂B and VP3 GH loops and alterations of viral genome structure as well as genome–capsid interactions during the uncoating. A low resolution cryo-electron tomography (cryo-ET) study[39] on poliovirus observed ~50-Å-length "umbilical" densities, putatively formed by VP1 N-terminus and VP4, which are used to deliver viral genome to the cytoplasm of the cell in an RNase-protected RNA transfer mechanism. The low resolution cryo-ET analysis and our high resolution structural information complement each other, further detailing the dynamic process for viral uncoating. Although we do not know the signal that triggers RNA release, we can now see that prior to this the genome decompresses itself to present a relatively loose, but multiple-layered structure, facilitating the egress of the genome, presumably via the size-limited channel at the twofold axes.

HFMD outbreaks used to be primarily caused by EV71 and CVA16. However, recent epidemics have been more dominated by CVA10, CVA6, and other HEV-A species[2,40]. Furthermore, frequent natural recombination events between HEV-A species[4] are a hindrance to accurately predicting the roles enteroviral subtypes might play during future outbreaks. Even though the EV71 vaccine on the market efficiently inhibits HFMD caused by EV71, it is not able to cross-protect from infections caused by other enteroviruses[5]. In addition, as EV71 vaccine becomes more widely used, the circulation pattern of HEV-As will continue to change. The HEV-A species can be subdivided into three groups based on sequence similarity, pathogenesis and receptor utilization: SCARB2-dependent subgroup, such as EV71 and CVA16; KREMEN1-dependent subgroup, including CVA10 and CVA6; X-receptor-dependent, e.g., EV76[15]. In this study, we report the mature virus structure for the KREMEN1-dependent subgroup of HEV-A viruses, providing implications for receptor binding; and have captured the dynamics in CVA10 uncoating, suggesting an ordered signal transmission mechanism for enterovirus uncoating, in which the canyon converts exogenetic receptor-attachment inputs to a generic RNA release mechanism. Systematic structural comparisons and comprehensive immunological studies of HEV-As will provide guidance for the rational design of novel and effective multivalent and broad-spectrum vaccine for protection against infections caused by EVs.

## Methods

**Virus production and purification.** Clinical specimens were collected from patients with CVA10 infection in Zhejiang Province, China. A ~0.5 mL of throat swab samples were inoculated into Vero cells for 1 h at 37 °C, then the samples were removed, and Dulbecco's modified Eagle's medium (DMEM, Sigma-Aldrich) supplemented with 2% fetal bovine serum (FBS, Gibco) was added. Cytopathic effect (CPE) was initially weak and increased CPE was observed over five passages. Vero cells were grown to 90% confluence and infected with CVA10 (GenBank accession no. AJK93551.1) at a multiplicity of infection (MOI) of 0.5. At 3 days post infection, the cultures were collected, resuspended in phosphate buffer saline (PBS) containing 1% NP-40 and lysed by three rounds of freezing and thawing. After lysis, the solution was centrifuged at 1,500 × g at 4 °C for 15 min to remove large debris. A discontinuous 20 and 50% sucrose gradient (wv⁻¹ in PBS) was used to purify viral particles. Fractions containing the CVA10 particles were pooled, concentrated, loaded onto a continuous 15 to 45% (wv⁻¹ in PBS) sucrose density gradient and centrifuged at 80,000 × g for 4 h. Three sets of fractions were collected and dialyzed against PBS buffer. Particles from these fractions were imaged by negative staining electron microscopy and cryo-EM analysis.

**Cryo-EM and data collection.** For cryo-grids preparation, 3 μL aliquot of purified CVA10 particles containing mature viruses, empty- and A- particles, (~3 mg mL⁻¹) were deposited onto fresh glow-discharged 400-mesh holey carbon-coated copper grids (C-flat, CF-2/1–2C, Protochips). Grids were blotted for 3.5 s in 80% relative humidity for plunge-freezing (Vitrobot, FEI) in liquid ethane. Cryo-EM data sets were collected at 300 kV with a Tian Krios microscope (FEI). Movies (25 frames, each 0.2 s, total dose of 25e⁻ Å⁻²) were acquired using a K2 detector with a defocus range of 0.8 to 2.3 μm. Automated single-particle data acquisition was performed by SerialEM, with a calibrated magnification of 37,037 × yielding a pixel size of 0.675 Å at super-resolution mode. The final images were binned, resulting in a pixel size of 1.35 Å for further data processing.

**Image processing.** In all, 1200 micrographs were recorded for the mixture of the CVA10 mature virus, empty- and A-particles. Out of these, 800 micrographs with visible CTF rings beyond 1/5 Å in their spectra were selected for further processing and ~150 micrographs produce high spatial frequencies beyond 3 Å, very close to its sampling limit (Nyquist frequency) (Supplementary Fig. 2c). The defocus value for each micrograph was determined using Gctf[41]. Particles were picked automatically by ETHAN[42] and false positives were manually removed using the boxer program in EMAN[43]. Gctf was used to estimate the contrast transfer function (CTF) parameters for drift corrected micrographs, and micrographs with significant astigmatism or drift were discarded. A total of 4835 CVA10 mature virions, 22,568 A-particles and 25,683 CVA10 empty particles were picked for the two-dimensional alignment and three-dimensional reconstruction using Relion[19] with recommended gold-standard refinement procedures[20] and applying icosahedral symmetry. The initial model for CVA10 was generated by low-pass filtering the reconstructed Aichivirus structure[44] to 30 Å. Two rounds of two-dimensional classification and three-dimensional refinement were performed to further select the particles for final refinement. The final resolution was evaluated using the gold-standard Fourier shell correlation (threshold = 0.143 criterion)[20]. We used the post processing in Relion[19] to perform the map sharpening (estimate the B-factor automatically) and the local resolution was evaluated by ResMap[45]. The data sets and refinement statistics are summarized in Supplementary Table 1.

**Model building and refinement.** The atomic models of CVA10 capsid proteins VP1, VP2, VP3, and VP4 were manually built into the density map in COOT[21] and refined with Phenix[46]. The crystal structures of CVA16 full particle[13] and uncoating intermediate[12] (PDB code 4JGZ and 5C4W) were used as homology models and fit into the cryo-EM map of CVA10. These complete atomic models were further improved in a pseudo-crystallographic manner by iterative positional and B-factor refinement in real space using Phenix[46] and rebuilt in COOT[21] against the cryo-EM maps. Densities for individual proteins were segmented, put in artificial crystal lattices, and then used to calculate their structural factors. The amplitudes and phases of these structural factors were used as pseudo experimental diffraction data for model refinement in Phenix[46]. The quality of the final model was confirmed visually by analyzing the match between map densities and coordinates and by calculation of correlation coefficients and R-factors. Refinement statistics are provided in Supplementary Table 1, as evaluated by Molprobity[47] functions integrated in Phenix[46].

**Symmetry-mismatch reconstruction.** For an icosahedral virus, its capsid is icosahedral but its genome has no symmetry. The 60 equivalent orientations of the icosahedral capsid can be easily obtained and the asymmetric genome structure may have a fixed orientation related to the symmetric capsid. The genome structure

within the capsid was reconstructed by using the symmetry-mismatch reconstruction method[35,48,49]. Firstly, we calculated a two-dimensional (2D) projection with CTF modulation for each raw particle image from the three-dimensional (3D) density map of capsid in its appropriate orientation. Then, we obtained the genome images by subtracting the 2D projections from the raw particle images. For the initial genome model generation, we randomly chose one of the 60 equivalent orientations of its capsid and reconstructed an intact particle structure (including the capsid and genome) using the raw particle images and obtained an initial model of the genome by masking the capsid structure. Lastly, we searched the asymmetric orientations of each genome image from the 60 equivalent orientations of its icosahedral capsid. The genome structure was determined by iterative orientation refinement and 3D asymmetric reconstruction. To validate the asymmetric reconstruction, we repeated this procedure by using a Gaussian ball to replace the above initial model, which gave the same result, confirming the asymmetric reconstruction was not influenced by the choice of initial model.

**Immunization of mice and in vitro neutralization assay.** The purified CVA10 mature viruses, empty- and A-particles were inactivated by formaldehyde (1:2000 dilution) at 25 °C for 3 days and their immunogenicities were evaluated in mice. Four groups of adult (4-weeks-old) BALB/c mice ($n = 6$ per group, three male and three female) were immunized (two doses, 2 weeks apart) with aluminum adjuvant (control), CVA10 mature viruses, empty- and A-particles, respectively. All animal procedures were carried out in accordance with the guideline for the Use of Animals in Research issued by the Institute of Biophysics, Chinese Academy of Sciences. Two weeks after the last immunization, sera samples were obtained and tested for ELISA and neutralization assays. Vero cell monolayers were diluted in DMEM supplemented with 2% FBS and then seeded into 96-well plates (~10,000 cells per well). Different concentrations of sera were diluted in DMEM by twofold serial dilutions ranging from 1:8 to 1:4096, and each well was incubated in 1:1 volume ratio with infectious CVA10 (100 TCID$_{50}$) for 1 h at 37 °C. Each sample was then incubated with the prepared Vero cells in 96-well plates at 37 °C and was inspected for CPE in infected cells for 3 days. The neutralization titers were the averages of the triplicates calculated based on the highest dilution in over 50% CPE.

**Analytical ultracentrifugation.** Sedimentation velocity experiments were performed on a Beckman XL-I analytical ultracentrifuge at 20 °C. Samples containing the three types of particles were diluted with PBS buffer (pH 7.4) to a proper concentration with an A280nm absorption of ~0.3, then loaded into a conventional double-sector quartz cell and mounted in a Beckman four-hole An-60 Ti rotor. Data were collected at 12,000 rpm at a wavelength of 280 nm. The SEDFIT software program was used to calculate interference sedimentation coefficients (www.analyticalultracentrifugation.com).

**Thermal stability assay.** An MX3005p reverse transcription polymerase chain reaction (RT-PCR) instrument (Agilent) was used to perform PaSTRy[50] experiments with the SYTO9 (Invitrogen) dye, which can detect the presence of RNA. We set up 50-μL reactions in a thin-walled PCR plate (Agilent), containing ~1 μL of either CVA10 mature virus or empty- or A-particles and 5 mM SYTO9 in PBS (pH 7.4), and the temperature ramped from 25 to 99 °C. Fluorescence was recorded in triplicate at 1 °C intervals.

## Data availability
The atomic coordinates of CVA10 mature virus, empty- and A-particles have been submitted to Protein Data Bank with accession numbers PDB: 6AKS, 6AKU, and 6AKT respectively. Cryo-EM density maps of CVA10 mature virus, empty particle, A-particle and viral genome have been deposited with the Electron Microscopy Data Bank: EMD-9642, EMD-9644, EMD-9643, and EMD-9652 respectively. A reporting summary for this Article is available as a Supplementary Information file. Other data are available from the corresponding authors upon reasonable request.

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

## Acknowledgements

We thank Prof. D. Stuart for providing the comments on the paper, Dr. Xiaojun Huang, Dr. Boling Zhu, and Dr. Zhenxi Guo for cryo-EM data collection at the Center for Biological imaging (CBI), Institute of Biophysics. Work was supported by the Strategic Priority Research Program (XDB29010000, XDB08020200), the National Key Research and Development Program (2014CB542800 and 2017YFC0840300), National Natural Science Foundation of China (31800145, 31770186, 31570742, 91530321, and 81520108019) and Technology Planning Project of Hunan Province (No. 2017RS3033). Yanjun Zhang was supported by Health leading Talents Program of Zhejiang Province. Xiangxi Wang was supported by Young Elite scientist sponsorship by CAST and the program C of "One Hundred of Talented People" of the Chinese Academy of Sciences.

## Author contributions

L.Z., Y.S., J.F., and L.C. performed experiments, Q.G., Y.Z. provided the reagents, L.Z., X.W., B.Z., and H.L. solved the structure, L.Z. and X.W. designed the study, all authors analyzed data, L.Z., Z.R., and X.W. wrote the manuscript.

## Additional information

**Competing interests:** The authors declare no competing interests.

