## [Peer Review File · Nature Communications]

Reviewers' comments:

Reviewer #1 (Remarks to the Author):

This manuscript describes the structure of the Coxsackie A10 virion, A particle, and empty capsid at near atomic resolution, and a lower resolution asymmetric reconstruction of the interior in the A particle showing semi-icosahedral density features that represent ordered segments of the RNA genome. CAV10 is a member of the A species of enteroviruses and is an important cause of hand-foot-and-mouth disease and less frequently causes more serious diseases in humans. Near atomic resolution structures of several other A species enterovirus including the EV71 virion and empty particle, the Coxsackie16 virion and A particle and the Coxsackie A10 provirion and A particle have been previously reported but this is the first structure in this group that uses KREMEN1 as a receptor. Not surprisingly, the overall structures are very similar, but there are several differences between the CAV10 structures and their counterparts in CAV6, CAV16 and EV71, notably a raised BC loop of VP1 and tilted EF loop of VP2 that alter the shape of the "canyon" and that the authors speculate could account for the utilization of KREMEN1 as a receptor, a difference in the structure of VP4 that results in this protein spanning two fivefold related protomers, and significantly more disorder in the GH loop of VP1 and EF loops of VP2 in the A particle than is seen in the other members of this group. This is also the first report of a asymmetric reconstruction of the viral genome in this group. The paper is generally well written, with some technical caveats listed below, the structural work is of very high quality, and the figures are of very high quality and help the reader understand the points the authors raise about the structure. The manuscript would be improved if the authors address the following points:

Technical points:

- 1) I might be missing something because the FSC plots are convincing, the representative densities shown are consistent with the claimed resolution, and the reports from the data base are supportive of the authors claims, but the fact that the reconstructions of the two expanded particles are quoted as 2.8Å and 2.7Å when the pixel size used for the reconstruction was 1.35Å raises a bit of a red flag as the resolution of the reconstructions are either very, very near or at the Nyquist limit, where one would expect the reconstructions to be degraded by interpolation error and perhaps even Fourier aliasing. The authors should address this issue as it will surely be noted by other readers.
- 2) The method used to generate the reconstruction of the RNA genome is very prone to biasing by the initial model. Has this reconstruction been done several times from different starting models?

Other points:

- 3) The source of the virus used was a patient sample that has been passage in Vero cells. If the information is available, it would be useful to virologists to have some idea of the passage history of the virus to know the number of generations of passage separate the virus used from that obtained from the patient.
- 4) The description "erection of the BC loop of VP1" in line 155 is perhaps unfortunate, and should be "raised conformation of the BC loop of VP1" (using the authors initial description).
- 5) The possibility that the first 61 aa of VP1 are not seen in the reconstruction of the A particle due to proteolysis (lines 231-2) seems unlikely and should be easily testable experimentally.
- 6) The statement that the genome organization of picornaviruses has proved difficult to visualize (line 257) is a bit misleading. There are now several picornaviruses (e.g. human parechovirus, Ljungan virus, and Seneca Valley virus where significant portions of the RNA genome are icosahedrally ordered and visible in structures, and there are numerous reports of significant

density for the viral genome in icosahedrally constrained structures at low to intermediate resolution. This should be noted. That said the use of symmetry breaking approaches to address genome structure is an important addition to this paper and is expected to become increasingly common in the future.

7) It would be informative to compare the structures of the asymmetric reconstruction of CAV10 with the icosahedrally ordered structures in parecho, Ljungan, and Seneca Valley viruses, and worth comparing the protein RNA contacts seen in CAV10 with those reported for these viruses and several additional viruses e.g CB3.

Reviewer #2 (Remarks to the Author):

The manuscript of Zhu et al describes the high resolution structure of the Coxsackie virus A10 (CVA10), a human type-A Enterovirus (HEV-A) which can cause serious clinical symptoms, including meningitis. As no vaccines or antiviral are available, there is increasing interest in understanding CVA10 in detail.

The study reports the structure of the native virus, as well of an entry-intermediate: the A-particle, and the final empty capsid. The comparison with other closely related enteroviruses, such as CVA16 and EV71, highlights structural differences which can explain why CVA10 uses a different receptor (Kremen1) than other HEV-A viruses which use SCARBA2 as receptor. Also, the three structures reported allow Zhu et al to propose a model for the early stages of HEV uncoating. As a bonus of the study, the authors report a that in the A-particle the genomic RNA is reordered into a defined structure; despite the low resolution of this structure, the finding is of extreme interest. The authors also show that the A-particle and the native virus have different antigenic properties; this underscores the importance of avoiding the conversion from native to the A-particle during vaccine production.

The manuscript is of adequate technical quality and it is well constructed. The findings are solid and advance our structural knowledge of picornaviruses in general and enteroviruses in particular. Such mechanistic details are essential for successful development of therapies, such a viral inhibitors and vaccines.

Comments:

- The authors have used cryo-EM to demonstrate the presence of three types of particles in their purified virus: the native virus, the A-particle, and the final empty particle. This characterisation is based exclusively on structural data without any biochemical argumentation. While there is an expected similarity between CVA10 and other enteroviruses, a more careful justification for the proposed capsid classification is desirable. Striking is the small number of native capsids (less than 10% of the total number) purified when compared to the A-particles and the empty capsids. The authors should speculate on possible causes of this observation.
- The presence, position, and nature of the hydrophobic pocket factor is an important subject for enteroviruses. The authors should offer a supplemental figure, or a separate panel showing the electron density of the CVA10 pocket and the density observed for the pocket factor. Also, a comparison with the A-particle as well as with CVA16 and EV71 could be informative.
- A supplemental figure with sequence alignment from CVA10, CV106, and EV71 to show the conserved regions and position of the loops discussed in the manuscript would make the reading more fluent.
- Despite the moderate resolution, the authors should discuss shortly how their finding compare with other picornaviruses for which an ordered RNA was reported, such as parechoviruses and Senecavirus. Does the native capsid shows any sign of ordered genome? Does the genome get ordered after the conversion to the A-particle? Did the authors observe something similar in their work on other picornaviruses?
- The asymmetric genome reconstruction is very informative. Nevertheless, we suggest two extra images (they could be supplementary) to make the finding more compelling. First, the authors

should show that the densities observed in the outer layer could indeed accommodate double stranded RNA; for instance a portion of a dsRNA fitted in one of the tubular densities. Second, a figure with the angular distribution of the views contributing to the final reconstruction should be shown, in order to prove that there is no bias in a certain views.

Minor points:

The manuscript needs a very careful grammar revision.

Line 61: should be 'Introduction'

Line 84: there should be a comma after to initiate infection

Line 130: this information is redundant with the methods section

Line 131: The sentence "The structures of ..." should be expanded for clarity.

Line 138: manually build using in COOT ?

Line 149: For clarity, the 'star-shaped' features should be indicated in a figure

Line 172: unclear, probably should be "inner surface surrounding this channel".

Line 250: it should be "in line with the..."

Line 332, 336, 417: it should be a space between 37 and °C.

Line 337: The sentence: "Used post processing in relation to evaluate the resolution" does not make sense. Probably the ResMap use should also be mentioned here.

Line 617: Figure C looks somehow confusing: dark blue, red and green worms represent N-termini and yellow the VP4. But we can see light blue coloured worms as well. A more clear colour scheme should be used..

Line 772: Figure S5 is somehow redundant, as the same information was presented in the main manuscript. Maybe the authors could use a zoom-in of the boxed area and show the individual residues

Response to referees' comments

We thank the reviewers for their positive and constructive comments, and we believe the paper is now improved.

Reviewers' comments:

Reviewer #1 (Remarks to the Author):

This manuscript describes the structure of the Coxsackie A10 virion, A particle, and empty capsid at near atomic resolution, and a lower resolution asymmetric reconstruction of the interior in the A particle showing semi-icosahedral density features that represent ordered segments of the RNA genome. CAV10 is a member of the A species of enteroviruses and is an important cause of hand-foot-and-mouth disease and less frequently causes more serious diseases in humans. Near atomic resolution structures of several other A species enterovirus including the EV71 virion and empty particle, the Coxsackie16 virion and A particle and the Coxsackie A10 provirion and A particle have been previously reported but this is the first structure in this group that uses KREMEN1 as a receptor. Not surprisingly, the overall structures are very similar, but there are several differences between the CAV10 structures and their counterparts in CAV6, CAV16 and EV71, notably a raised BC loop of VP1 and tilted EF loop of VP2 that alter the shape of the “canyon” and that the authors speculate could account for the utilization of KREMEN1 as a receptor, a difference in the structure of VP4 that results in this protein spanning two fivefold related protomers, and significantly more disorder in the GH loop of VP1 and EF loops of VP2 in the A particle than is seen in the other members of this group. This is also the first report of a asymmetric reconstruction of the viral genome in this group. The paper is generally well written, with some technical caveats listed below, the structural work is of very high quality, and the figures are of very high quality and help the reader understands the points the authors raise about the structure. The manuscript would be improved if the authors address the following points:

We thank the reviewer for a high evaluation of our manuscript and considering it an important contribution.

Technical points:

1) I might be missing something because the FSC plots are convincing, the representative densities shown are consistent with the claimed resolution, and the reports from the data base are supportive of the authors claims, but the fact that the

reconstructions of the two expanded particles are quoted as 2.8Å and 2.7Å when the pixel size used for the reconstruction was 1.35Å raises a bit of a red flag as the resolution of the reconstructions are either very, very near or at the Nyquist limit, where one would expect the reconstructions to be degraded by interpolation error and perhaps even Fourier aliasing. The authors should address this issue as it will surely be noted by other readers.

Thanks for pointing this out and sorry that we did not make this key point as clearly as we should have done. Indeed, the resolution of our reconstructions is very near to the Nyquist limit, which might benefit from the following two reasons. 1), The data sets were collected with a final pixel size of 0.675 Å at super-resolution mode, then these images were binned resulting in a pixel size of 1.35 Å for further data processing. 2), Out of these data sets, ~800 micrographs with visible CTF rings beyond 1/5 Å in their spectra were selected for further processing and ~150 micrographs produce high spatial frequencies beyond 3 Å, very close to its sampling limit (Nyquist frequency). All these details as well as related figures (Fig. S2b-S2c) have been added in the revised manuscript.

2) The method used to generate the reconstruction of the RNA genome is very prone to biasing by the initial model. Has this reconstruction been done several times from different starting models?

Thanks for your suggestion. We have provided more details on the reconstruction of viral genome in the revised version. Agreeably, the reconstruction of the RNA genome is very prone to biasing by the initial model. To overcome this problem, we used two different initial models: a map obtained by masking the capsid structure and a Gaussian ball to refine and reconstruct the genome structure, both yielded the same result.

Other points:

3) The source of the virus used was a patient sample that has been passage in Vero cells. If the information is available, it would be useful to virologists to have some idea of the passage history of the virus to know the number of generations of passage separate the virus used from that obtained from the patient.

Thanks for your suggestion. We have added the passage history of this virus in the method section. -- As follows “Clinical specimens were collected from patients with CVA10 infection in Zhejiang Province, China. A ~0.5 mL of throat swab samples were inoculated into Vero cells for 1h at 37 °C, then the samples were removed, and Dulbecco’s modified Eagle’s medium (DMEM, Sigma-Aldrich) supplemented with 2% fetal bovine serum (FBS, Gibco) was added. Cytopathic effect (CPE) was initially weak and increased CPE was observed over five passages”

4) The description “erection of the BC loop of VP1” in line 155 is perhaps unfortunate, and should be “raised conformation of the BC loop of VP1” (using the authors initial description).

Thanks, corrected!

5) The possibility that the first 61 aa of VP1 are not seen in the reconstruction of the A particle due to proteolysis (lines 231-2) seems unlikely and should be easily testable experimentally.

Thanks for your suggestions. We did a protein composition analysis (provided in Fig. S1) and found the first 61 residues of VP1 are not cleaved. Therefore, we have modified the statement- as follows “the first 61 residues of VP1 are disordered”.

6) The statement that the genome organization of picornaviruses has proved difficult to visualize (line 257) is a bit misleading. There are now several picornaviruses (e.g. human parechovirus, Ljungan virus, and Seneca Valley virus where significant portions of the RNA genome are icosahedrally ordered and visible in structures, and there are numerous reports of significant density for the viral genome in icosahedrally constrained structures at low to intermediate resolution. This should be noted. That said the use of symmetry breaking approaches to address genome structure is an important addition to this paper and is expected to become increasingly common in the future.

Thanks for pointing this out. We have removed the sentence “the genome organization of picornaviruses has proved difficult to visualize”. As you suggested, brief discussions on the application of the asymmetric reconstruction in viral genome structure determination have been added. –As follows “Although multiple X-ray crystallographic and icosahedrally reconstructed cryo-EM structures of picornaviruses show small portions of the viral RNA, given the fact that the bulk of the genome does not exhibit icosahedral symmetry, reconstruction with the imposition of icosahedral symmetry would average the densities, obscuring the genome reconstruction. To overcome this problem, we reconstructed the capsid and genome using the symmetry mismatch reconstruction method (without imposing any symmetry), being expected to become a common strategy to study asymmetric structural features in icosahedral viruses.”

7) It would be informative to compare the structures of the asymmetric reconstruction of CAV10 with the icosahedrally ordered structures in parecho, Ljungan, and Seneca Valley viruses, and worth comparing the protein RNA contacts seen in CAV10 with those reported for these viruses and several additional viruses e.g CB3.

Thanks for your suggestions. Structural comparisons of the asymmetric reconstruction of CVA10 with the icosahedrally ordered structures in parecho,

Ljungan, and Seneca Valley viruses have been provided in Fig. S10 in the revised manuscript.

Reviewer #2 (Remarks to the Author):

NCOMMS-18-04918-T

The architecture of the Herpes simplex virus C-capsid

Reviewer #2 (Remarks to the Author):

The manuscript of Zhu et al describes the high resolution structure of the Coxsackie virus A10 (CVA10), a human type-A Enterovirus (HEV-A) which can cause serious clinical symptoms, including meningitis. As no vaccines or antiviral are available, there is increasing interest in understanding CVA10 in detail.

The study reports the structure of the native virus, as well of an entry-intermediate: the A-particle, and the final empty capsid. The comparison with other closely related enteroviruses, such as CVA16 and EV71, highlights structural differences which can explain why CVA10 uses a different receptor (Kremen1) than other HEV-A viruses which use SCARBA2 as receptor. Also, the three structures reported allow Zhu et al to propose a model for the early stages of HEV uncoating. As a bonus of the study, the authors report a that in the A-particle the genomic RNA is reordered into a defined structure; despite the low resolution of this structure, the finding is of extreme interest. The authors also show that the A-particle and the native virus have different antigenic properties; this underscores the importance of avoiding the conversion from native to the A-particle during vaccine production.

The manuscript is of adequate technical quality and it is well constructed. The findings are solid and advance our structural knowledge of picornaviruses in general and enteroviruses in particular. Such mechanistic details are essential for successful development of therapies, such a viral inhibitors and vaccines.

We thank the reviewer for a high evaluation of our manuscript and considering it an important contribution in picornaviruses.

Comments:

- The authors have used cryo-EM to demonstrate the presence of three types of particles in their purified virus: the native virus, the A-particle, and the final empty particle. This characterisation is based exclusively on structural data without any biochemical argumentation. While there is an expected similarity between CVA10 and other enteroviruses, a more careful justification for the proposed capsid classification is desirable. Striking is the small number of native capsids (less than 10% of the total number) purified when compared to the A-particles and the empty capsids. The authors should speculate on possible causes of this observation.

Thanks for your suggestions. We agree! The characterizations of these three types of particles, including protein composition analysis, sedimentation coefficients determination and RNA detection have been added in supplementary information (in Fig. S1). Regarding the ratios of the three particles, we have also added some discussions on the possible causes of this observation in the revised manuscript as follows- “For many enteroviruses, two (mature viruses and empty particles) or three (mature viruses, A- and empty particles) predominant types of viral particles with various ratios are produced during a natural infection and the ratios of these particles can be influenced by a couple of factors, including virus strain, multiplicity of infection, cell culture conditions and cell types. Our results provide lessons for CVA10 vaccine development: the procedures for virus production need to be optimized to decrease amounts of antigenically altered empty particles, which are likely to dilute the useful portion of a vaccine”.

- The presence, position, and nature of the hydrophobic pocket factor is an important subject for enteroviruses. The authors should offer a supplemental figure, or a separate panel showing the electron density of the CVA10 pocket and the density observed for the pocket factor. Also, a comparison with the A-particle as well as with CVA16 and EV71 could be informative.

Thanks for your suggestions. The densities of the CVA10 pocket and its pocket factor have been provided in Fig. S6a. Structural comparisons of the hydrophobic pockets of three CVA10 particles with CVA16 and EV71 have been provided in Fig. S6b and discussed in the revised version, as follows—“As expected, both expanded particles lack visible VP4 and harbor neither an hydrophobic pocket nor pocket factor in VP1 due to the movements of the polypeptide backbone in residues 229–233 at the end of the GH loop and the start of strand H, and the conformational changes of the Met229 and Phe232 side chains”.

- A supplemental figure with sequence alignment from CVA10, CV106, and EV71 to show the conserved regions and position of the loops discussed in the manuscript would make the reading more fluent.

Thanks, sequence alignments from CVA10, CVA6, CVA16 and EV71 have been provided in Fig. S4.

- Despite the moderate resolution, the authors should discuss shortly how their finding compare with other picornaviruses for which an ordered RNA was reported, such as parechoviruses and Senecavirus. Does the native capsid shows any sign of ordered genome? Does the genome get ordered after the conversion to the A-particle? Did the authors observe something similar in their work on other picornaviruses?

Thanks for pointing this out. Structural comparisons of the asymmetric reconstruction of CVA10 A-particle with the icosahedrally ordered structures in parecho, Ljungan,

and Seneca Valley viruses have been provided in Fig. S10 and discussed in the revised manuscript. Interestingly, like other enteroviruses, the CVA10 mature virion shows no notable signs of ordered genome, while the genome becomes ordered as the conversion to the A-particle and similar observations have been reported in CVA16 uncoating intermediate at very low resolution (revealed by X-ray crystallography, Ren et.al Nat Commun, 2013).

- The asymmetric genome reconstruction is very informative. Nevertheless, we suggest two extra images (they could be supplementary) to make the finding more compelling. First, the authors should show that the densities observed in the outer layer could indeed accommodate double stranded RNA; for instance a portion of a dsRNA fitted in one of the tubular densities. Second, a figure with the angular distribution of the views contributing to the final reconstruction should be shown, in order to prove that there is no bias in a certain views.

Thanks for your suggestions. Fitting of a double stranded RNA fragment to the genome densities and angular distribution analysis for the final asymmetrical reconstruction have been provided in Fig. S9.

Minor points:

The manuscript needs a very careful grammar revision.

Line 61: should be 'Introduction'

Thanks, done!

Line 84: there should be a comma after to initiate infection

Thanks, done!

Line 130: this information is redundant with the methods section

Thanks, done!

Line 131: The sentence "The structures of ..." should be expanded for clarity.

Thanks, we have modified the statement – as follows "A total of 4,586, 22,725, and 21,456 particles were used to reconstruct the structures of CVA10 mature virus, empty- and A-particles with icosahedral symmetry imposed by single-particle techniques using Relion. The final resolutions of maps for CVA10 mature virus, empty- and A-particles are 3.0, 2.7, and 2.8 Å, respectively using the "gold" standard Fourier shell correlation (FSC) = 0.143 criterion".

Line 138: manually build using in COOT ?

Thanks, corrected!

Line 149: For clarity, the 'star-shaped' features should be indicated in a figure

Thanks, done! The "star-shaped" features have been marked by black lines in Fig. 2a.

Line 172: unclear, probably should be "inner surface surrounding this channel".

Thanks, corrected!

Line 250: it should be "in line with the..."

Thanks, done!

Line 332, 336, 417: it should be a space between 37 and °C.

Thanks, done!

Line 337: The sentence: "Used post processing in Relion to evaluate the resolution" does not make sense. Probably the ResMap use should also be mentioned here.

Thanks, we have modified the statement -- as follows "We used the post processing in Relion to perform the map sharpening (estimate the B-factor automatically) and the local resolution was evaluated by ResMap".

Line 617: Figure C looks somehow confusing: dark blue, red and green worms represent N-termini and yellow the VP4. But we can see light blue coloured worms as well. A more clear colour scheme should be used..

Sorry that we did not describe the color scheme clearly in the Figure legend. We have added more descriptions in the revised version. --As follows "Rainbow ribbon representations of the VP4 structures (blue N terminus through green and yellow to red C terminus) in CVA10, CVA16 and EV71 mature viruses. VP1 (light blue), VP2 (light green) and VP3 (salmon) structures are rendered as surfaces".

Line 772: Figure S5 is somehow redundant, as the same information was presented in the main manuscript. Maybe the authors could use a zoom-in of the boxed area and show the individual residues.

Thanks, done!

REVIEWERS' COMMENTS:

Reviewer #1 (Remarks to the Author):

I have reviewed the revised manuscript and supplementary material and the author's response to the reviewers comments on the original submission. I feel that the authors have done a very good job of addressing each of the comments from the two reviews and that the manuscript is now suitable for publication.

Reviewer #2 (Remarks to the Author):

I consider that the authors have answered adequately all the points raised in the reviews. The manuscript is now much improved and - in my opinion - ready for publication.

Very minor point : I would prefer that each supplementary figure to be on a single page (not a page plus a couple of lines).